# A physiologically-based model of localized mucociliary clearance in the airways

Monica E. Shapiro[1], Timothy E. Corcoran [1,2,3¤*], Carol A. Bertrand[4], Robert S. Parker[1,3,5]

**1** Department of Chemical & Petroleum Engineering, University of Pittsburgh, Pittsburgh, Pennsylvania, United States of America, **2** Division of Pulmonary, Allergy, Critical Care, and Sleep Medicine, University of Pittsburgh, Pittsburgh, Pennsylvania, United States of America, **3** Department of Bioengineering, University of Pittsburgh, Pittsburgh, Pennsylvania, United States of America, **4** Department of Pediatrics, University of Pittsburgh, Pittsburgh, Pennsylvania, United States of America, **5** McGowan Institute for Regenerative Medicine University of Pittsburgh, Pittsburgh, Pennsylvania, United States of America

¤ Current address: UPMC MUH NW628, 3459 Fifth Ave, Pittsburgh, Pennsylvania, 15213, United States of America
* tec23@pitt.edu

## Abstract

The mucociliary clearance (MC) system clears mucus, pathogens, and toxins from the airways. Whole lung MC rate can be measured using gamma camera imaging after the inhalation of radiolabeled particulate. We sought a means to evaluate the therapeutic effect of clearance enhancing therapies in different airway size groups. We developed a mathematical model of mucus transport in the right lung that, when informed by imaging data, estimates MC rate and unclearable activity at points across the airway tree. We fit the model to imaging studies from 11 healthy controls (HC), resulting in a per-point mean absolute error (MAE) of 0.085±0.016% of the total particulate deposition. Using principal component analysis and hierarchical clustering, we reduced the number of fitted clearance rate coefficients from 114 to 5 with only an 8.7% increase in MAE. These 5 cluster groups were closely associated with specific regions of the lung and likely with specific airway size groups. Comparing the HC group to a cystic fibrosis (CF) group we found only one cluster with significantly depressed MC rates in CF corresponding to the lower lobe. The inhalation of 7% hypertonic saline (HS) by the CF group increased MC rate in all clusters and decreased unclearable activity in 4/5 clusters. The computational model described provides detailed regional estimates of MC rate when applied to clearance imaging studies. If further informed, this model may provide a valuable tool for studying small airways obstructive disease and evaluating mucus clearance-enhancing therapies in the lung.

## Introduction

The mucociliary clearance (MC) system is a vital host defense that protects the lungs and upper airways from inhaled pathogens and toxins. It includes a protective

**Data availability statement:** The raw data has been deposited in GitHub. The link is https://github.com/monshap/mcc-grid-data.

**Funding:** Funding through NIH - National Heart, Lung, and Blood Institute, 1 UO1 HL131046-01.

**Competing interests:** No authors have competing interests.

layer of mucus that lines the airways and traps inhaled bacteria, viruses, and toxins. Beneath the mucus layer is a watery periciliary layer that contains the airway cilia. The cilia beat synchronously with their tips propelling the mucus from smaller airways towards larger airways and out of the lungs. Failure of the MC system, as occurs in genetic diseases such as primary ciliary dyskinesia (PCD), results in mucus obstruction, chronic airway infection, and may increase the effects inhaled toxins, including cigarette smoke. MC defects have been described in CF [1–4], asthma [5], COPD [6], and Covid-19 [7]. MC rate can be measured using a technique where non-absorbable radiolabeled particles are inhaled in a liquid aerosol prior to the collection of sequential planar gamma images over time – an MC scan. This technique has been used to evaluate the efficacy of clearance enhancing therapies [8–10], and is useful for measuring whole lung clearance, but it is less able to detect local clearance defects or defects in specific airway size groups. The measurement can also be affected by the initial distribution of the radiolabeled particles into the lung [2] which can vary with lung disease.

Previous work has sought to determine more detail on MC rates in specific airway size classes by dividing the planar scintigraphy images into different regions of interest (ROIs) and tracking the change in activity in each of those regions. Most commonly, a rectangular "central" region that is half the height and width of the whole lung and centered at the medial edge and a "peripheral" region containing the remaining lung area have been used [4]. The peripheral region contains small airways (approximately generation 6 and higher) and alveoli, while the central region would contain these as well as all the large airways (generations 1–5 starting from the trachea). Previous work has proposed the use of an adjusted measure of whole lung MC rate calculated using the percent of aerosol deposited in the central ROI relative to the study average [11]. While this can provide more even comparisons of MC within a given study, it relies on a very coarse measurement of initial particle distribution.

Radioactive counts at any given point in the airway tree are affected by local clearance efficacy and the rate at which particulate is being transported in by more distal airways. A mathematical model of mucociliary clearance that describes the transport of radiolabeled particulate throughout the lung would allow for the detection of local defects in clearance and allow characterization of MC rate in specific airway size classes. It would also remove the confounding effects caused by differences in initial particulate distribution. Previous work from our group includes a dynamic model of MC rate and airway surface liquid absorption at the population level [3]. The benefit of this type of modeling is that the parameters derived from the model are not dependent on the deposition pattern of the aerosol and therefore can be compared across individuals to assess MC rate. The model from Markovetz et al. [3] was based on the previously described central and peripheral ROIs and was adequate for describing the dynamics observed at the population level. When applied to individuals, however, the model did not consistently describe the dynamics observed, particularly in the peripheral region. This was caused by: 1) the model assuming activity within the ROIs was evenly distributed, which is visibly not the case for most subjects, and

2) assuming the clearance from the peripheral ROI into the central ROI was negligible, which was done to improve the identifiability of other model parameters. At the population level, the whole lung clearance was relatively insensitive to peripheral-to-central clearance. However, this caused rather large errors when applied to specific individuals, which limits the model's utility for characterizing individualized MC rates. Therefore, we sought a more detailed model that would better capture individualized response and peripheral-to-central MC while maintaining parameter identifiability.

The remainder of this paper discusses the development of our mucus transport model and its application to study of MC rate in healthy people and people with cystic fibrosis (CF) under baseline and therapeutic conditions [11]. Inhaled hypertonic saline (HS, 7%) is frequently used to treat CF. It creates an osmotic gradient on the airway surface that improves hydration and transportability of the mucus, thus improving MC [12]. Isotonic saline (IS, 0.9%) typically achieves a lesser increase in MC than HS [13] and was used in the study as a comparator. To the best of our knowledge, this is the first mathematical model to accurately capture spatial differences in individual MC dynamics across the lungs.

## Materials and methods

### Nuclear imaging studies

The healthy controls (HC) used for developing this model, as well as the cystic fibrosis (CF) group included in the study, were drawn from a larger overall study [11], and their demographics are summarized in Table 1. Participants inhaled two radiolabeled probes, a non-absorbable particle, Technetium-99m labeled sulfur colloid (Tc-SC) and a small molecule, Indium-111 labeled diethylene triamine pentaacetic acid (In-DTPA) prior to sequential gamma camera imaging. We focused here on Tc-SC, which provided a surrogate to measure MC rate. (In-DTPA provided an additional assessment of paracellular absorption.) Tc-SC and In-DTPA were delivered together in a liquid aerosol using a breathing pattern that preferentially targeted aerosol delivery to the airways rather than the alveoli. Two-minute planar images of the lungs were then continually collected for 80 minutes while the patient lay supine. After 10 minutes of imaging, all subjects inhaled an aerosolized intervention of either IS or HS while remaining supine while the images were continually being collected. All subjects had a day where they inhaled IS, but only CF subjects also had a study day where they inhaled HS, and the order of these study days was randomized. Further details of the study procedure can be found in [11]. This study was approved by the Institutional Review Board of the University of Pittsburgh (STUDY19030274). Written informed consent was obtained from all participants. Participant recruitment occurred between January 23, 2017 and November 6, 2019.

**Table 1. Demographics of included study participants with lung clearance index (LCI) and pulmonary function data (mean ± standard deviation). For corrector/potentiators in CF group, I = ivacaftor, L/I = lumacaftor/ivacaftor, and T/I = tezacaftor/ivacaftor. The study preceded the availability of elexacaftor. LCI (lung clearance index) is a multi-breath gas washout technique that provides a measure of peripheral airway obstruction. FEV1 = one-second forced expiratory volume % of predicted. FVC = forced vital capacity % of predicted. FEF$_{25-75}$ = forced expiratory flowrate between 25 and 75% of FVC, % of predicted. IS = isotonic saline (0.9%). HS = hypertonic saline (7%).**

| | | Controls (HC) | Cystic Fibrosis (CF) | | |
|---|---|---|---|---|---|
| Number of Subjects | | 11 | 23 | | |
| Age (years) | | 22 ± 3 | 30 ± 14 | | |
| Female/Male | | 4/7 | 13/10 | | |
| Corrector/potentiator Drugs in use | | N/A | I | L/I or T/I | None |
| | | | 4 | 7 | 11 |
| LCI | | 7.3 ± 0.89 (n = 10) | 11 ± 2.9 (n = 20) | | |
| | | | IS Day | HS Day | |
| Pulmonary Function Tests (% predicted) | FEV$_1$ | 102 ± 12 | 69 ± 25 | 69 ± 25 | |
| | FVC | 106 ± 13 | 84 ± 22 | 80 ± 25 | |
| | FEF$_{25-75}$ | 88 ± 20 | 48 ± 36 | 48 ± 33 | |

## Image processing

Decay correction of the two radioisotopes used was performed within the gamma camera. The resulting images were then processed using ImageJ (NIH) to manually obtain a matrix of pixel intensity values (which corresponded to radioactive counts associated with particulate) for the anterior and posterior images overlaying the right lung. (Only the right lung was analyzed to avoid interference from the stomach.) Whole lung regions of interest were selected by scaling an anatomically-based whole lung ROI from Alcoforado et al. [14] to the right lung of the posterior transmission scan for each subject in ImageJ. This outline was then aligned to the right lung Tc-SC activity of the posterior and horizontally-mirrored for the anterior images. Pixel intensities within the outline were extracted and exported for further processing in Python.

In Python, corrections for background activity were applied and then the images were divided into a grid of 16 rows and 8 columns with equal pixel areas for an individual. This grid shape was chosen to enable direct comparison with traditional central and peripheral ROIs. (By convention, the central lung ROI typically is defined as a rectangle with ½ the height and the width of a rectangle containing the whole lung region.) The geometric mean between the posterior and anterior views of corresponding images was used to minimize the effect of MC into and out of the plane of imaging. Note that the image processing procedures here were different from those used in our previous publication on these subject groups [11] which did not incorporate geometric mean correction.

## Model construction

Firstly, we sought to define the compartments in our model so that the activity within them was uniform. The central and peripheral regions used in Markovetz et al. [3] are commonly applied in MC scans, however, they only provide a coarse division between large and small airways. In reality, both regions contain small airways and alveoli. Work has been done to develop more physiologically relevant ROIs. Tossici-Bolt et al. [15] used concentric whole lung outlines to divide the area into regions equidistant from the trachea. While this showed good numerical agreement with predicted deposition, the spatial distribution of activity within these regions was very heterogeneous and performed poorly when used as a basis for a compartmental model. Similarly, Alcoforado et al. [14] developed anatomically based central and peripheral ROIs based on averaged HRCT scans. While deposition in their central region was more uniform than the previous version, the deposition within their peripheral region was still very heterogeneous, and consequently the model also performed poorly. Other groups have looked at very localized (pixel-by-pixel) differences in particulate activity to account for this heterogeneity [16]. In the absence of a model, however, it was unclear how much the change in activity was due to mucus clearance from that pixel and how much was due to mucus flowing into it from other pixels. Additionally, slight mismatches between frames could introduce substantial noise if pixels were used as a basis for the compartments. Instead, we opted for a moderate resolution approach. The 16 x 8 grid used to define our compartments contained at least 5 x 5 pixels in each grid. We then derived flow constraints between these grids based on anatomical averages.

To accurately track MC dynamics, we also needed to account for particulate that did not clear because of where it deposited in the lung. A breathing pattern designed to maximize aerosol delivery to the airways was used, but likely some particulate still deposited in alveoli. This particulate was unlikely to clear in the timeframe of this study. In the case of CF, there may also have been airway regions of collapsed cilia, such that particulate that landed on these regions did not clear by the end of the study. We refer to this as "non-clearable activity" which we estimated as the lower 25% quartile of activity for each grid over the course of the study. This method was chosen rather than just using the final time points because some grids had an increase in activity over the course of study due to mucus flowing in from other grids over the 80-minute study session. The non-clearable activity was subtracted and only the remaining, "clearable" activity was considered in the dynamic model.

The airway tree is branched, such that mucus clears from many smaller, less ciliated airways into larger, more ciliated airways, which feed into larger and larger airways until it is cleared up the trachea and swallowed. To implement this

behavior as flow constraints, we considered grids with visibly distinct airways separately from those which were too small to distinguish. We defined grids with visible airways as the large airway region (LAR), based on anatomically averaged high resolution computed tomography scans from Greenblatt et al. [17]. The right lung from that work was divided into the same 16 x 8 grid and the LAR defined as any grid that overlapped the visible large airways. Within this region, flow constraints were defined, such that mucus could only flow to other grids in the LAR that were closer to or equidistant from the trachea. Outside this region, flow was constrained to clear only towards this LAR. These flow constraints were implemented by assigning each grid an elevation level, as shown in Fig 1, where elevations less than or equal to 5 corresponded to the LAR. Mucus could only flow to adjacent grids (including diagonals) and could not flow to grids at higher elevations than their own. We assumed that mucus flowed out of a grid element at equal rates in all permitted directions.

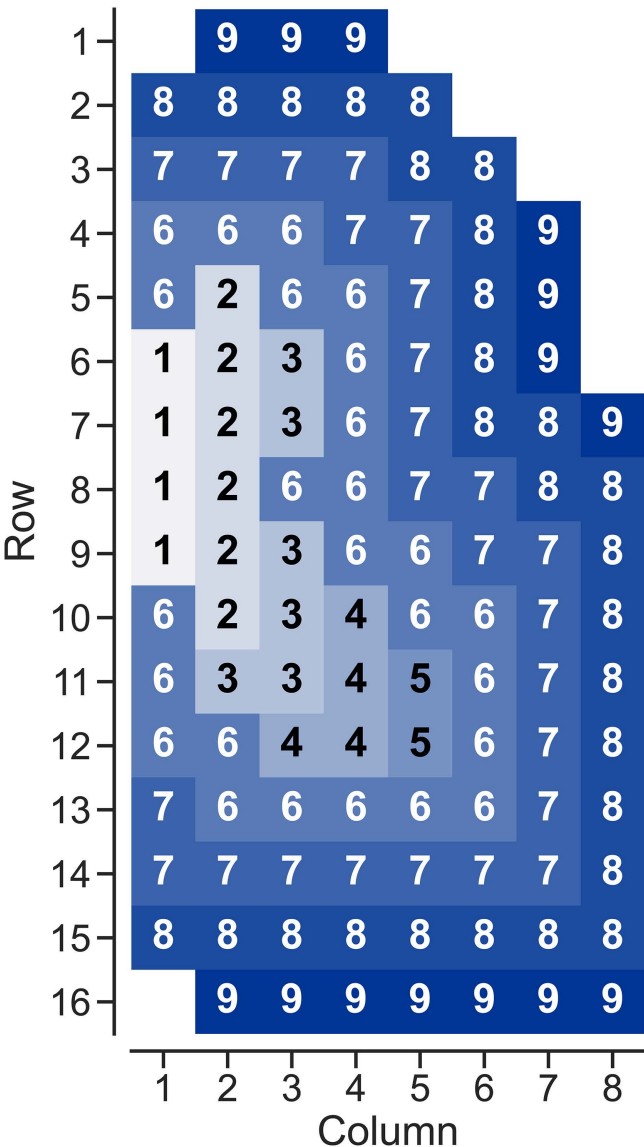

**Fig 1. Elevation map defining directions where mucus can be transported.** At elevations 3 and lower, mucus can flow to grids of lower or equal elevation. At elevations 4 or greater, mucus can only flow to lower elevations. All mucus flow rates from a given grid cell are the same.

Using these flow constraints, a material balance of the radiolabeled particulate around each grid in the network was used to derive ordinary differential equations describing the change in concentration of particulate in each grid over time. We assumed that MC rate was constant over the course of a study and that within a grid, the particulate was well-mixed. We also assumed that net flow into and out of the imaging plane was negligible when considering the geometric mean of the anterior and posterior images. Because we were only considering motion in the plane of imaging, we treated concentration as activity per unit area, rather than volume. Thus, the concentration of particulate in the $i$th row and $j$th column ($\hat{C}_{i,j}(t)$) could be described using Equation (1), where $N$ was the set of coordinates (other (row, column) combinations) of grids that could flow into grid ($i,j$) and $X_{i,j}$ was the number of grid cells that particulate in grid ($i,j$) could flow out to. Using row 3, column 1 as an example, $N$ would be the set [(2, 1), (2, 2)] of all grids that could flow into grid (3, 1) and $X_{i,j}$ would be 2 since grid (3, 1) could flow into grids (4, 1) and (4, 2).

$$\frac{d\hat{C}_{i,j}}{dt} = \sum_{n \in N} k_n \hat{C}_n(t) - X_{i,j} k_{i,j} \hat{C}_{i,j}(t)$$

(1)

The resulting network of ordinary differential equations contained 114 free parameters, corresponding to the clearance rate coefficients ($k_{i,j}$) from each grid. These parameters were fit simultaneously by minimizing the sum (over space) of the sum of squared residuals (over time) between model and data using nonlinear least-squares optimization as expressed in Equation (2). Here $C_{i,j}(t)$ was the clearable activity calculated from the nuclear images and $\hat{C}_{i,j}(t)$ was the model estimate in grid ($i,j$), at time $t$.

$$\min_{k_{i,j}} \sum_{i=1}^{16} \sum_{j=1}^{8} \sum_{t} \left( C_{i,j}(t) - \hat{C}_{i,j}(t) \right)^2$$

(2)

## Model reduction

To address the issue of parametric identifiability, we took steps to reduce the number of fitted parameters in our model while maintaining the physiological relevance derived from flow constraints. To reduce parameter flexibility in the more distal regions, where there were many grids at equal elevations, we considered threshold elevations at or above which mucus could only flow to lower elevations. We evaluated the change in error across healthy controls using each elevation as a threshold and looked for an elbow. This indicated a cut-off point where flow to equal elevations was necessary to describe the dynamics for at least one individual. Since airways of similar size should have comparable MC rates, we were interested in fitting similar grids that were spatially connected with a single clearance rate coefficient.

To generate such grid clusters, we used constrained hierarchical clustering – a tool with several applications in image segmentation [18,19]. In traditional hierarchical clustering, a similarity metric is used to determine how similar two grids or clusters are to one another, and then the algorithm iteratively merges the most similar pair. Here we used Ward's method [20], which minimized the variance of the merged pairs. Connectivity constraints were implemented by assigning similarities of zero to unconnected grids. This method assumed linear independence of the features – in this instance across our subjects. Particularly in healthy controls, where we would not expect any plugged airways, we would expect similar patterns of dynamic behavior in most of the controls, violating this assumption. For this reason, we first transformed our parameter space using an eigenvalue decomposition technique called principal component analysis (PCA).

In PCA, the covariance matrix of feature observations is broken into orthogonal eigenvectors, referred to as principal components (PCs), with corresponding eigenvalues to form a basis set for the covariance matrix. An eigenvalue thus represents the total variance captured in the direction of a PC and by dividing each eigenvalue by the sum of all the eigenvalues, we can calculate the proportion of variance attributed to each PC. By using only a subset of these PCs with high eigenvalues, much of the original information is retained, but with a reduced number of features [21]. This technique

has been successfully used for image denoising [22] and facial recognition [23], since most of the low variance directions discarded are noise. In this case, we applied PCA to the fitted parameters in each grid, rather than to raw pixel intensities. Since some participants had non-identifiable parameters for some of the grids, this method had the added benefit of minimizing the effect these parameters had on clustering.

Combining these methods, new clearance rate coefficients were fit – one per cluster – to each individual, such that the original ODEs remained the same, but all clearance rate coefficients within a cluster were set to the same value. We then evaluated the change in error associated with fitting a single clearance rate coefficient to each cluster and selected the fewest number of clusters that did not dramatically increase the change in error for our final model.

## Results

### Full scale model

The mass-action kinetic model described in the Methods section was fit to MC data from 11 healthy controls. The trajectories for three of these individuals are shown in Fig 2 along with model simulations. The 114 model parameters were fit simultaneously to data from all grids. Model simulations fit the dynamic trends observed in each grid, as well as those observed at the whole lung scale, and had a per-point mean absolute error (MAE) of 0.085±0.016% of the total particulate deposition. Although only the healthy controls were used for model development and reduction, the model also adequately captures dynamics observed in CF individuals, with a per-point MAE of 0.21±0.059% of total particulate deposited. This is illustrated in Fig 3, where trajectories and model simulations for three CF individuals are shown for the same grids and in the whole lung. Additionally, it highlights how very different dynamics across individual grids can give rise to apparently similar whole lung measurements.

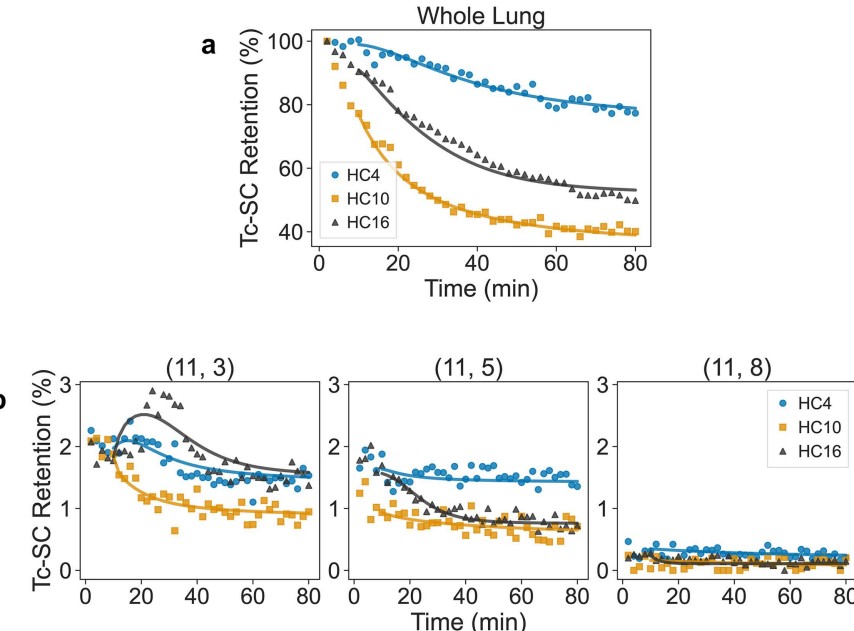

**Fig 2. Comparison between fitted model (solid lines) and data (markers) at the whole lung scale (a) and in three example grids from row 11, columns 3, 5, and 8, which are at different elevations (b).** Colors and markers correspond to three individual healthy subjects (HC4, 10, and 16). Radioactivity is normalized by starting counts after decay and background correction and reported on a percentage basis as Tc-SC retention. Grid depicted in Fig 1.

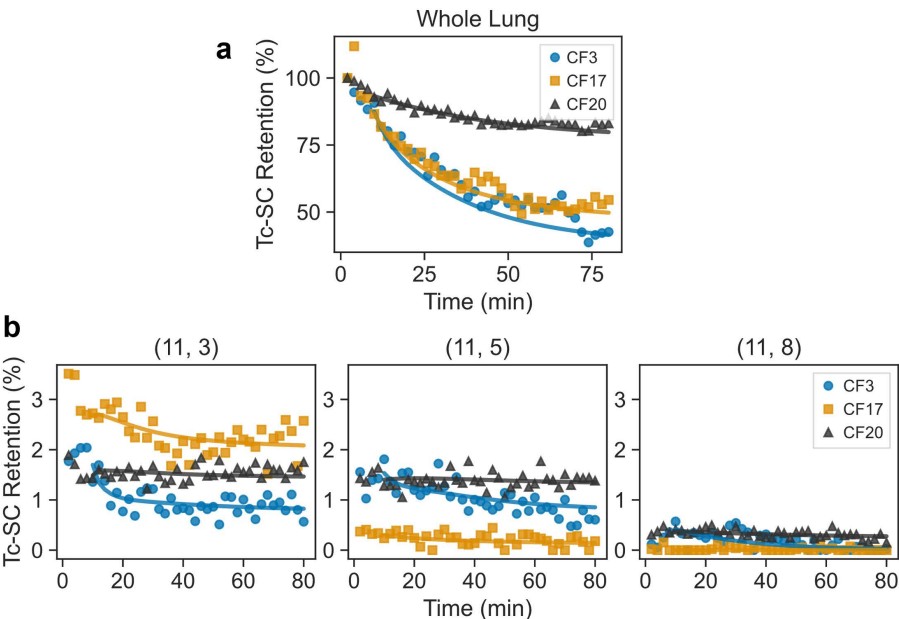

Fig 3. **Comparison between fitted model (solid lines) and data (markers) at the whole lung scale (a) and in the same three example grids as** Fig 2b. Colors and markers correspond to three individual CF subjects (CF3, 17, and 20). Radioactivity is normalized by starting counts after decay and background correction and reported on a percentage basis as Tc-SC retention. Grid depicted in Fig 1.

## Model reduction

To reduce model flexibility, especially in the distal regions, adjustments to the flow constraints were considered based on their elevations. In an iterative process working from highest to lowest elevation, the flow of particles from grids at or above a threshold elevation was only allowed to flow to lower elevations and not to grids with equal elevation. The change in total sum of squared error (across all time points and grids) between model and data was calculated for each of these threshold elevations relative to the full-scale model, as shown in Fig 4, which captures the trade-off between decreasing

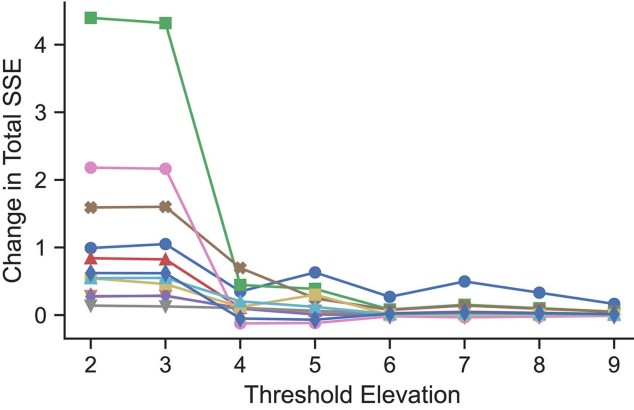

Fig 4. **Change from full-scale to reduced model in sum (across all time points and grids) of squared error between simulation and MC scan data as a function of threshold elevation.** Mucus could only flow to equal elevation grids below the threshold elevation. Each line represents a healthy control.

model flexibility (by only allowing flow to lower elevations) and increasing model error. There was a noticeable increase in relative error between elevations 4 and 3, indicating that flow between grids at elevation 3 was necessary to accurately describe the MC dynamics for some participants. We therefore constrained flow in grids with elevation 4 or higher to only flow to lower elevations but allowed flow at equal elevation for grids with elevations 3 or less for the remainder of the model reduction.

The parameters of the healthy controls were distributed approximately log-normally. Since PCA is most effective when performed on features of zero mean and equal variance, the parameter distributions for each subject were transformed by taking the natural logarithm and scaling it to the standard normal distribution. After performing PCA on this transformed distribution, we discarded PCs that explained <5% of the variance across participants. This left only the first two PCs, which combined captured 74% of the variance across all healthy subjects, as shown in Fig 5. These PCs are represented spatially in Fig 5, where red indicates the positive direction and blue represents the negative direction and the closer in color two grids are, the more similar the parameter values are in those grids across the healthy participants. This can be conceptualized as new axes which describe the parameters for each individual. The first PC approximated how different MC rates were in the large vs small airways for a participant, and the second approximated how different these rates were between upper and lower lobes. The MC rates for each grid and each participant could therefore be described (to within 74% of the original variance) as a linear combination of just these two PCs, rather than the original 114 MC rates fitted for each individual. By using only the first two principal components, we ensured the variables used for clustering were linearly independent and reduced the impact of non-identifiable parameters by removing PCs that had little variance across healthy controls.

Hierarchical clustering was performed on the grids in the model using these first two PCs, as shown in Fig 6a. The dendrogram in Fig 6a depicts the most similar connected clusters or grids from these PCs merging from right to left, where each row represents an individual grid. To determine an appropriate number of clusters to use for a reduced model, the number of clusters was increased (starting from 2 clusters) by splitting the grids at each vertical line working from left to right. A single clearance rate coefficient was fit for each of the clusters and the change in error between the full and reduced model was calculated at each split, as shown in Fig 7. There was a large decrease in error when using 5 clusters vs 4 ($16.7 \pm 9.96$ vs $26.3 \pm 15.2$), respectively), so we selected the 5 clusters shown in Fig 6 as our final model. Despite this drastic decrease in the number of fitted parameters (114 to 5), the average per-point MAE was only $0.092 \pm 0.016\%$ of total initial deposition, representing an 8.7% increase from the original model.

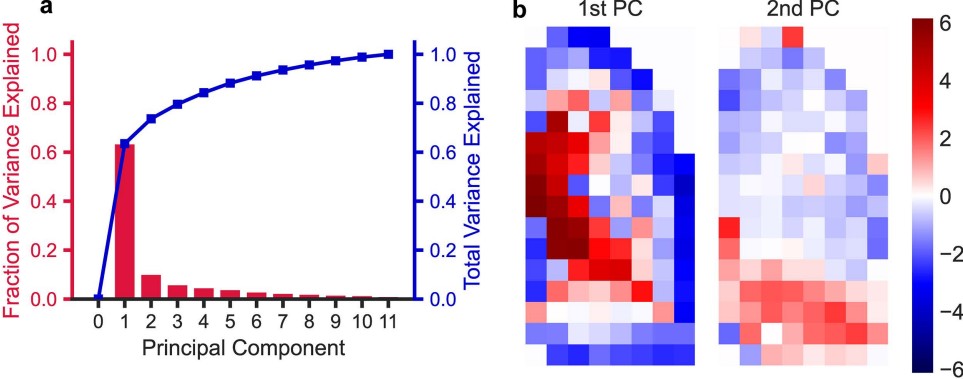

**Fig 5. Application of model using two principal components.** a) Principal component (PC) analysis of log-normalized clearance rate coefficients across healthy subjects. PCs are orthogonal and ordered by the amount of variance in the data they explain. b) The first two PCs, represented spatially. The scale bar on the right indicates the magnitude and direction of the components from the median. Parameters of all healthy subjects can be described as a linear combination of these two PCs to within 74% of the original variance.

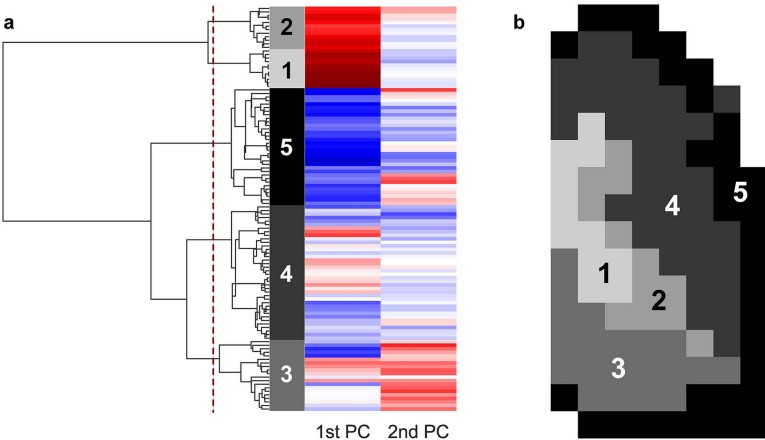

**Fig 6. Hierarchical clustering.** a) Dendrogram and heatmap showing agglomerative clustering of 1st and 2nd principal components of healthy subject clearance rate coefficients and b) spatial layout of selected clusters. Each colored row of the heatmap represents a grid in terms of principal components (columns). Each vertical line on the dendrogram represents the merging of the next most similar cluster or grid. The dashed line shows where the dendrogram was truncated to form the selected clusters.

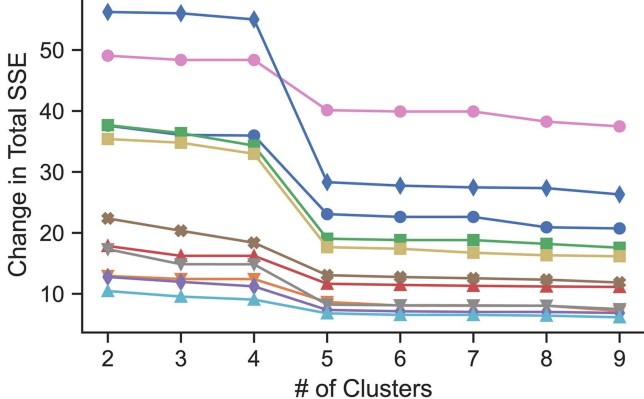

**Fig 7. Change from full-scale to reduced model in sum (across all time points and grids) of squared error between model and MC scan data as a function of the number of clusters used in the reduced models.** Each line represents a healthy control.

## Model application to assess therapeutic effect

CF is a genetic condition that results in the absence or dysfunction of a specific anion channel (CFTR: the cystic fibrosis transmembrane conductance regulator) on epithelial surfaces including the airway epithelium. CFTR transports ions which generate osmotic gradients that contribute to normal hydration of the airway mucus. The absence of CFTR causes mucus dehydration, failed MC, airway obstruction, opportunistic infection, and inflammation. Inhaled (nebulized) isotonic and hypertonic saline solutions are used to treat CF. They contribute to osmotic gradients in the airways that help to hydrate the mucus resulting in effective MC.

Clearance rate coefficients for the reduced model were fit for three subgroups, as summarized in Table 1: healthy controls (HC) after inhaled IS, CF subjects after inhaled IS (CFIS), and CF subjects after inhaled HS (CFHS). The distribution of parameters for each subgroup and cluster are shown in Fig 8. Groups were compared using two-sample Kolmogorov-Smirnov non-parametric tests for HC vs CFIS. CFIS vs CFHS were compared using a paired Wilcoxon

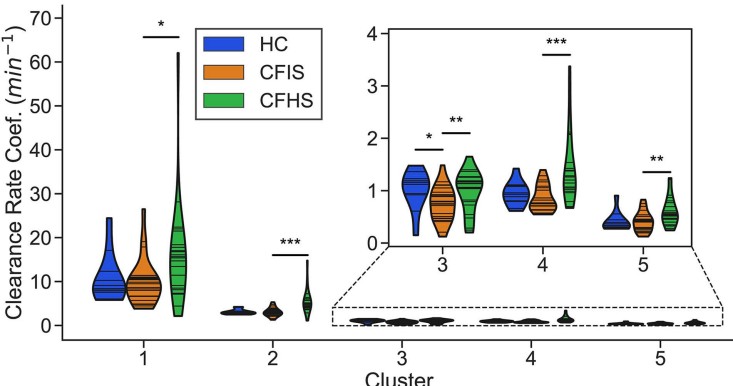

**Fig 8. Clearance rate coefficient distribution by subgroup (HC = healthy controls including inhalation of isotonic saline (IS), CFIS = CF including inhalation of IS, CFHS = CF including inhalation of 7% hypertonic saline (HS)) and cluster.** CFIS and CFHS are paired data from the same subjects on two different study days. For each violin, the width of the shaded region shows the estimated probability density of the corresponding parameter values; each horizontal line corresponds to the fitted parameter value for an individual. HC vs CFIS were compared using a two-sample Kolmogorov-Smirnov statistical test and CFIS vs CFHS were compared using a paired Wilcoxon signed-rank test (*: p < 0.05, **: p < 0.01, ***: p < 0.001).

signed-rank test. There was only one cluster that had significantly different parameters between HC and CFIS, [# 3] which approximately overlays the lower lobe. In contrast, when comparing CFIS and CFHS parameters, all clusters were significantly different. In fact, many of the clearance rate coefficients in the CFHS group were elevated beyond what was observed in the HC group. Additionally, the magnitude of clearance rate coefficients in clusters 1 and 2, which overlay the large airways, are an order of magnitude greater than those of the other clusters. Fig 9 includes the distributions of unclearable activity by cluster group. Unclearable activity was significantly increased in CF in group #5 only (peripheral lung). Hypertonic saline decreased unclearable activity in 4/ 5 cluster groups, not including the peripheral lung.

## Discussion

The model developed provides a new tool to assess imaging studies of mucociliary clearance in the airways that has better spatial resolution than previous methods and can capture a broad range of individuals. Unlike previous methods, which divided the lung into two lumped regions, this model captured mucus transport in 2D across the entire lung. This

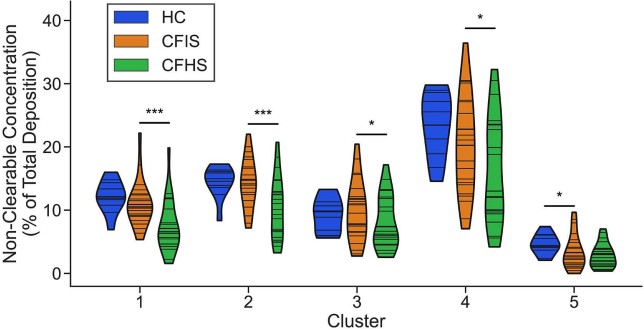

**Fig 9. Distribution of non-clearable activity by subgroup (HC = healthy controls including inhalation of isotonic saline (IS), CFIS = CF including inhalation of IS, CFHS = CF including inhalation of hypertonic saline (HS)) and cluster.** CFIS and CFHS are paired data from the same subjects on two different study days. For each violin, the width of the shaded region shows the estimated probability density of the corresponding non-clearable activity; each horizontal line corresponds to the value for an individual. HC vs CFIS were compared using a two-sample Kolmogorov-Smirnov statistical test and CFIS vs CFHS were compared using a paired Wilcoxon signed-rank test (*: p < 0.05, **: p < 0.01, ***: p < 0.001).

enabled the description of more complex dynamics by accounting for transit time from the peripheral small airways to the distal large airways. Models that lump the entire peripheral region together cannot capture this phenomenon. Additionally, by tracking mucus flow both into and out of the airways, this model provided a better depiction of how mucus clears than traditional endpoint measurements.

By increasing the resolution of our model, we identified 5 key regions in the right lung that could be used to capture the mucus clearance characteristics of all individuals in our study, including those of the CF group, who were not included in the original clustering process. These clusters were based on neighboring grids with similar clearance rates. Interestingly, the clusters that emerged from our analysis are spatially consistent with anatomical features of the airway tree. Specifically, clusters 1 and 2, respectively, covered the mainstem bronchus and the bronchus intermedius. The increased clearance rate coefficients in these clusters are also consistent with observations that MC is faster in the large airways, which are more heavily ciliated [24]. Cluster 5 was made up of grids at the periphery of the lung, which contained mostly small airways and alveoli, and accordingly this cluster had a lower clearance rate coefficient than the other clusters. Finally, clusters 3 and 4 divided the remaining space into upper and lower regions, respectively. The variability in clearance rate coefficients between these two clusters could be explained by differences in clearance rate or airway length between different lobes of the lung. While it might be expected that there would be three distinct clusters corresponding to the three lobes in the right lung, due to the 2D nature of these studies, there would be substantial overlap between the middle and upper lobes that may explain the discrepancy.

We found only one region where there the distribution of clearance rate coefficients was significantly different between HC and CF groups, cluster 3, corresponding to the lower lobe. Previous work describing this data showed no significant difference in whole lung MC between CF and HC groups, when adjusted for percent central deposition [11]. This points to our model as possibly a more sensitive tool for detecting localized changes to MC. While previous studies have consistently found evidence of worse disease in the right vs left lung, the results are highly varied as to whether there is any difference in disease severity between lobes [25–30]. Although those studies did not specifically look at MC as a marker of regional disease severity, one would expect regions of depressed MC to accompany those with high degrees of bronchiectasis, air trapping, inflammatory markers, and bacterial loads. These results suggest that MC is slowed most in the middle or lower lobe in CF, which is consistent with findings of increased air trapping in the middle and lower lobes relative to the upper lobe in CF [28,31].

One reason we may not see significantly depressed MC in other regions may be related to how we processed the non-clearable activity. Since the clearance rate coefficients were only fit to mucus that cleared over the course of the 80-minute study window, it is possible that in regions with high degrees of mucus plugging, the radiolabeled aerosol simply got trapped and never cleared. If this were the case, one would expect to see a significant increase in the amount of non-clearable Tc-SC relative to the healthy controls, which we did not observe (see Fig 9). It is alternatively possible that due to decreased ventilation, no aerosol deposited in regions with very severe disease. Since all subjects had some clearable mucus observed over the course of the experiment in each cluster, however, we think the impact of this on our results was minimal. The inhalation of IS may have normalized MC rates in the CF group, potentially masking MC depression that would otherwise be apparent in this group.

Clearance rate coefficients increased significantly after inhaled HS in every cluster in the CF group. This is consistent with studies showing increases in whole lung MC in CF subjects following inhaled HS [11,12,32–36]. Interestingly, some of these coefficients exceeded the clearance coefficients observed in healthy subjects. While this would seem to suggest HS could enhance MC capabilities of CF subjects beyond those of healthy subjects, it is important to note that these coefficients only captured the speed of mucus clearance and not the duration of the effect.

Significant decreases in non-clearable activity were also observed between IS and HS study days in all but cluster 5 (the most peripheral region). This suggests that HS can acutely increase the total amount of mucus that can be cleared in addition to the speed of clearance. The difference was most pronounced in the regions overlapping the mainstem

bronchus and bronchus intermedius (clusters 1 and 2). The increase in clearable activity in clusters 3 and 4 was more moderate, suggesting that rehydration of mucus may have been less effective in these regions.

One substantial limitation to the application of our model is that the overall sample size was quite small. In particular, the PCs used for hierarchical clustering were derived from parameters for only 11 healthy subjects. Also, the group was on average young (22±3 years) and majority male (7/4). It is likely that with additional subjects, the PCs would be slightly different and therefore the ultimate clusters would vary slightly from our reduced model. Additionally, we selected a grid size of 16 x 8 so that we could easily compare our results to prior work that used a rectangular central region. It may be that a different grid size could better capture boundaries between different dynamic regions and result in higher quality fits or additional clusters that may not be distinguishable at the current grid size. This is an area that could be further explored in future studies. Despite that, these clusters were able to accurately capture dynamics from 57 study days total including both healthy and diseased individuals, so we would not expect the differences from our current model to be great. An additional limitation is that all participants in this study inhaled isotonic or hypertonic saline during the imaging procedure, so our "baseline" measurements of MC may therefore be accelerated relative to other studies. Finally, some error may have been incurred based on the use of camera-based decay correction which may not correctly account for background counts prior to correction. We estimate that these errors were relatively small are would not substantially affect the study outcomes.

External validation of this model would provide greater confidence in the conclusions drawn. Towards that end, we have provided a detailed procedure on how to extract the necessary information in ImageJ and have automated the processing of that data and model fitting in Python. All code and a tutorial for using it are available on GitHub (https://github.com/monshap/psanalysis). We hope that this will facilitate the application of this model to other studies, including study of other airway diseases.

In summary, we have developed a novel dynamic model of mucociliary clearance that is able to account for heterogeneous probe aerosol deposition patterns and the transport of mucus from more distal airways into more proximal ones. The model needs to be more completely informed with data from more subjects across a more complete range of ages. When fully informed, this model can be applied to clearance imaging data to gain insight into regional airway pathophysiology and therapeutic response.

## Author contributions

**Conceptualization:** Monica E. Shapiro, Timothy E. Corcoran, Carol A. Bertrand, Robert S. Parker.

**Formal analysis:** Monica E. Shapiro, Timothy E. Corcoran.

**Funding acquisition:** Timothy E. Corcoran.

**Investigation:** Timothy E. Corcoran, Carol A. Bertrand, Robert S. Parker.

**Methodology:** Timothy E. Corcoran, Carol A. Bertrand, Robert S. Parker.

**Project administration:** Timothy E. Corcoran.

**Resources:** Robert S. Parker.

**Software:** Monica E. Shapiro, Robert S. Parker.

**Supervision:** Carol A. Bertrand, Robert S. Parker.

**Validation:** Monica E. Shapiro.

**Visualization:** Monica E. Shapiro.

**Writing – original draft:** Monica E. Shapiro.

**Writing – review & editing:** Timothy E. Corcoran, Carol A. Bertrand, Robert S. Parker.

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
