## [Decision Letter · Decision Letter 0]

20 Jun 2025

Dear Dr. Corcoran,

Thank you for submitting your manuscript to PLOS ONE. After careful consideration, we feel that it has merit but does not fully meet PLOS ONE’s publication criteria as it currently stands. Therefore, we invite you to submit a revised version of the manuscript that addresses the points raised during the review process.

We look forward to receiving your revised manuscript.

Kind regards,

Saidul Islam, Ph.D.

Academic Editor

PLOS ONE

Journal Requirements:

3. In the online submission form, you indicated that [Insert text from online submission form here].

Reviewers' comments:

Reviewer's Responses to Questions

**Comments to the Author**

1. Is the manuscript technically sound, and do the data support the conclusions?

Reviewer #1: Yes

Reviewer #2: Yes

2. Has the statistical analysis been performed appropriately and rigorously?

Reviewer #1: Yes

Reviewer #2: I Don't Know

3. Have the authors made all data underlying the findings in their manuscript fully available?

Reviewer #1: Yes

Reviewer #2: Yes

4. Is the manuscript presented in an intelligible fashion and written in standard English?

Reviewer #1: Yes

Reviewer #2: Yes

Reviewer #1: The authors have developed a unique and novel dynamic model for the mucociliary clearance (MC) to account for heterogeneous probe aerosol deposition patterns and mucus transport.

Their future study for their model update to be more completely informed with data from more subjects at different age ranges could be a perfect fit to be published as a following paper in PLOS One.

Reviewer #2: This paper describes a novel mathematical method for analyzing the regional deposition, and more to the point, the regional mucociliary clearance of particles deposited in the human lung airways. Applied to pharmaceutical interventions in lung diseases, it can produce a more nuanced measure of clearance activity than the predominant current methods.

1. Page 2, line 34/35. Abstract - “significantly depressed MC rates in CF corresponding to the lower lobe.” Somewhere in the abstract should state this applies to right lung only since only that lung was analyzed.

2. Page 7, line 135. Image Processing- Was the scaling of standard Alcoforado ROIs done by subject height, lung volume, linear dimensions, CT scan? Would that affect pixel size or pixel count?

1. Page 7, line 135. Image Processing- Alcoforado ROIs are presumably done with normal HC, but it is known that CF, over-weights, dense breast tissue can have a different transmission scan profile. In addition, CT scans done supine in Alcoforado models may produce different lung shapes than MC image done recumbent. Could these affect the regional analysis? Or not by much?

2. Page 7, line 140. Image Processing- “corrections for background activity were applied at this point”. This was done after the camera software had done the isotope decay factoring. It has been my contention that the camera decay factor did not include a previous background subtraction, or used a preset background, therefore also decay correcting the background counts, which is incorrect. This calculation would have a disproportionate effect in areas of lower counts, i.e., peripheral, compared to areas of high counts such as large airways or “local hot spots”. Was this effect accounted for during your background subtraction? Or is it moot? Were local regional background counts subtracted or whole lung counts?

3. Page 7, line 147. Image Processing- “Note that the image processing procedures here were different from those used in our previous publication”. This is a creditable warning. This begs two questions though. 1. How is this procedure different from previous, its impact on the change and why the change, or 2. What is the reason for mentioning the previous procedure since it does not apply?

4. Page 8. Image Processing- ““non-clearable activity” which we estimated as the lower 25% quartile of activity…” This may be a completely valid approach, but the use of the value of 25% would benefit from a better explanation for determining this cutoff. Why not choose 5%, 23%, 42% etc.? How does 25% compare to an average value at the end of the study?

5. Page 12, line 260 and Figure 2- It would be beneficial to illustrate in a separate figure how the analysis fits to CF in order to compare/evaluate its relevance.

**Do you want your identity to be public for this peer review?** For information about this choice, including consent withdrawal, please see our Privacy Policy

Reviewer #1: **Yes: ** Tevfik Gemci, PhD

Reviewer #2: **Yes: ** Kirby L Zeman

---

## [Author Response · Author response to Decision Letter 1]

17 Sep 2025

Reviewer #1: The authors have developed a unique and novel dynamic model for the mucociliary clearance (MC) to account for heterogeneous probe aerosol deposition patterns and mucus transport. Their future study for their model update to be more completely informed with data from more subjects at different age ranges could be a perfect fit to be published as a following paper in PLOS One.

*We thank the reviewer for their comments. In the future we plan to more completely inform the model and would hope to include a range of ages.

Reviewer #2: This paper describes a novel mathematical method for analyzing the regional deposition, and more to the point, the regional mucociliary clearance of particles deposited in the human lung airways. Applied to pharmaceutical interventions in lung diseases, it can produce a more nuanced measure of clearance activity than the predominant current methods.

1. Page 2, line 34/35. Abstract - “significantly depressed MC rates in CF corresponding to the lower lobe.” Somewhere in the abstract should state this applies to right lung only since only that lung was analyzed.

*Added to the upper portion of the abstract. Line 26

2. Page 7, line 135. Image Processing- Was the scaling of standard Alcoforado ROIs done by subject height, lung volume, linear dimensions, CT scan? Would that affect pixel size or pixel count?

*The scaling of the ROI was done manually in ImageJ. There are no changes to the image itself made during that scaling so there should be any changes in pixel size. After scaling, pixel counts inside the ROI would be similar to what would be seen with a hand drawn lung outline. I’ve added some words to the methods to make this a bit clearer.

1. Page 7, line 135. Image Processing- Alcoforado ROIs are presumably done with normal HC, but it is known that CF, over-weights, dense breast tissue can have a different transmission scan profile. In addition, CT scans done supine in Alcoforado models may produce different lung shapes than MC image done recumbent. Could these affect the regional analysis? Or not by much?

*Everything in Alcoforado and in our work was done with participants supine, so that matches. (I’ve corrected our text which originally said recumbent which is less specific than supine.) I think the manual scaling was accurate enough that the standard outlines were well matched to the participant’s lung shape and thus that any differences in regional clearance caused by their use was minimal.

2. Page 7, line 140. Image Processing- “corrections for background activity were applied at this point”. This was done after the camera software had done the isotope decay factoring. It has been my contention that the camera decay factor did not include a previous background subtraction, or used a preset background, therefore also decay correcting the background counts, which is incorrect. This calculation would have a disproportionate effect in areas of lower counts, i.e., peripheral, compared to areas of high counts such as large airways or “local hot spots”. Was this effect accounted for during your background subtraction? Or is it moot? Were local regional background counts subtracted or whole lung counts?

*I suspect the reviewer is correct that there was not a uniform background subtraction performed as part of the machine’s decay correction and that the machine is therefore decay correcting noise and adding it to the image. As the reviewer is aware, the machine-borne software is somewhat of a black box. We had a single measurement for background that we subtracted from whole lung Technetium counts at all time points. I ran some numbers on this and it seems like the amount of error this would have caused is relatively minimal. The “the decay corrected background” would have amounted to about 2% of the image counts. We have added words to the discussion describing this limitation.

3. Page 7, line 147. Image Processing- “Note that the image processing procedures here were different from those used in our previous publication”. This is a creditable warning. This begs two questions though. 1. How is this procedure different from previous, its impact on the change and why the change, or 2. What is the reason for mentioning the previous procedure since it does not apply?

*The major difference is the use of anterior and posterior counts with geometric mean correction. We view this specific effort to be exploratory. It made sense to publish the original data using more conventional methods to allow comparisons to previous work. We’ve added some words to explain this major difference.

4. Page 8. Image Processing- ““non-clearable activity” which we estimated as the lower 25% quartile of activity…” This may be a completely valid approach, but the use of the value of 25% would benefit from a better explanation for determining this cutoff. Why not choose 5%, 23%, 42% etc.? How does 25% compare to an average value at the end of the study?

*We chose this value (which corresponds to 10 data points) as a balance between using sufficient data to mitigate noise in the measurements without losing data to inform the dynamic parameters of the model (the model states can never go below 0 and thus subtracting off too much “non-clearable activity” can result in negative data points that the model cannot fit). We compared using fewer and more data points to define “non-clearable activity” and found 10 points to provide a better balance across all individuals. In grids that have their lowest activity at the end of the study, these values are very comparable, but for grids where the activity increases during the study, the values can be quite different. If we just use the final measurements in those grids, the dynamics all occur at values <0 and thus cannot be fit by our model.

5. Page 12, line 260 and Figure 2- It would be beneficial to illustrate in a separate figure how the analysis fits to CF in order to compare/evaluate its relevance.

*We have added the suggested figure with model fits to CF individuals. While we did not use the CF individuals for model development, the new figure highlights the applicability of using this model in that population.

---

## [Editor Report · Decision Letter 1]

15 Oct 2025

A Physiologically-based Model of Localized Mucociliary Clearance in the Airways

PONE-D-25-07038R1

Dear Dr. Corcoran,

We’re pleased to inform you that your manuscript has been judged scientifically suitable for publication and will be formally accepted for publication once it meets all outstanding technical requirements.

Kind regards,

Saidul Islam, Ph.D.

Academic Editor

PLOS ONE
---

## [Editor Report · Acceptance letter]

PONE-D-25-07038R1

PLOS ONE

Dear Dr. Corcoran,

I'm pleased to inform you that your manuscript has been deemed suitable for publication in PLOS ONE. Congratulations! Your manuscript is now being handed over to our production team.

Kind regards,

on behalf of

Dr Saidul Islam

Academic Editor

PLOS ONE